# ReasoningLM: Enabling Structural Subgraph Reasoning in Pre-trained Language Models for Question Answering over Knowledge Graph

**Jinhao Jiang[1,3], Kun Zhou[2,3], Wayne Xin Zhao[1,3]***, **Yaliang Li[4], Ji-Rong Wen[1,2,3]**

[1]Gaoling School of Artificial Intelligence, Renmin University of China.
[2]School of Information, Renmin University of China.
[3]Beijing Key Laboratory of Big Data Management and Analysis Methods.
[4]Alibaba Group.
{jiangjinhao,jrwen}@ruc.edu.cn,francis_kun_zhou@163.com
yaliang.li@alibaba-inc.com, batmanfly@gmail.com

## Abstract

Question Answering over Knowledge Graph (KGQA) aims to seek answer entities for the natural language question from a large-scale Knowledge Graph (KG). To better perform reasoning on KG, recent work typically adopts a pre-trained language model (PLM) to model the question, and a graph neural network (GNN) based module to perform multi-hop reasoning on the KG. Despite the effectiveness, due to the divergence in model architecture, the PLM and GNN are not closely integrated, limiting the knowledge sharing and fine-grained feature interactions. To solve it, we aim to simplify the above two-module approach, and develop a more capable PLM that can directly support subgraph reasoning for KGQA, namely ReasoningLM. In our approach, we propose a subgraph-aware self-attention mechanism to imitate the GNN for performing structured reasoning, and also adopt an adaptation tuning strategy to adapt the model parameters with 20,000 subgraphs with synthesized questions. After adaptation, the PLM can be parameter-efficient fine-tuned on downstream tasks. Experiments show that ReasoningLM surpasses state-of-the-art models by a large margin, even with fewer updated parameters and less training data. Our codes and data are publicly available at https://github.com/RUCAIBox/ReasoningLM.

## 1 Introduction

Question answering over knowledge graph (KGQA) (Sun et al., 2018; He et al., 2021) has garnered significant attention in recent years, which aims to find answers for the natural language questions based on knowledge graphs (KGs), *e.g.,* Freebase (Bollacker et al., 2008) and Wikidata (Tanon et al., 2016). Since a massive amount of world knowledge has been formatted into a structured form (*e.g.,* a triple $\langle head, relation, tail \rangle$) in the KG, we can develop KGQA methods by leveraging structural semantics of KG to more accurately infer the answer entities to complex factual questions.

Starting from the topic entities mentioned in the question, a typical KGQA approach (Sun et al., 2018) is to perform the multi-hop reasoning along with relations on the KG, until finding a path that can reach the answer entities. To develop this approach, existing methods (Sun et al., 2018; He et al., 2021; Shi et al., 2021) mostly incorporate *a text encoder* to produce the representation of the given question, and *a reasoning module* to perform multi-hop reasoning on the KG using the question representation. Typically, recent work (Jiang et al., 2022b,a) adopts the pre-trained language models (PLM) (*e.g.,* BERT (Devlin et al., 2019)) and graph neural networks (GNN) (*e.g.,* GAT (Velickovic et al., 2017)) to implement the text encoder and reasoning module respectively, which can better understand the semantic information in the question and structured knowledge from the KG, improving the final performance

Despite the effectiveness, there are two major shortcomings with the aforementioned approach that combines a PLM encoder and a GNN reasoner. First, due to different model architectures, PLM and GNN are often integrated in a loose way (*e.g.,* relevance score sharing), which largely limits the knowledge sharing and fine-grained interaction between the question and KG (a subgraph spanned by related entities). Second, the GNN based reasoner performs reasoning mainly based on subgraph structure, which lack rich semantic knowledge as that in PLMs, making the reasoning results likely to be less effective, especially for complex questions. In addition to the two shortcomings, such a approach also requires a complicated implementation in practice, since it involves in two different modules.

To address these issues, we aim to simplify the above two-module approach and develop a more

---

* Corresponding author.

capable PLM that can directly support structural subgraph reasoning for KGQA. Our approach is inspired by the finding that the Transformer architecture consisting of stacked self-attention modules can be explained as a fully-connected graph encoder (Dwivedi and Bresson, 2020) in a mathematically equivalent way. Therefore, Transformer essentially has the potential to effectively model graph data and further performs graph reasoning, which has been shown in existing work (Ying et al., 2021). While, these attempts based on graph Transformers either neglect the modeling in text semantics, or cannot capture fine-grained semantic interaction between question and KG subgraph, making them infeasible for KGQA tasks.

To this end, in this paper, we propose a subgraph reasoning enhanced PLM, called *ReasoningLM*, enabling both effective question understanding and KG reasoning in a unified approach. As the major contribution, we propose a *subgraph-aware self-attention* mechanism, which can imitate the GNN to model the entities and their relations via attending to neighboring nodes on the KG. Further, such a structural attention mechanism has been integrated into a constrained masking framework, to jointly model question attention, KG to question attention, and KG attention. In this way, we can not only perform the knowledge interaction and sharing between the question and KG, but also leverage PLM to perform structural reasoning as GNN. However, since the PLM is originally trained by learning from general-purpose natural language text, it is necessary to adapt it to the special input format and attention mechanism. Thus, we propose an *adaptation tuning* strategy that utilizes 20,000 subgraphs with synthesized questions to adapt the parameters of the PLM. After adaptation, the PLM has been well adapted for subgraph reasoning, hence it can be fine-tuned on different downstream KGQA tasks in a parameter-efficient manner, achieving better performance with only a few parameters trained.

To evaluate the effectiveness of our approach, we conduct extensive experiments on three KGQA datasets. Experimental results demonstrate that our proposed approach can surpass existing state-of-the-art models by a large margin, even with fewer updated parameters and less training data.

Our contributions can be summarized as follows:

• We enable the PLM to simultaneously model question understanding, deep interaction between question and subgraph, and reasoning over sub-graph by leveraging an adapted subgraph-aware self-attention mechanism.

• We propose an automatic data construction method for the KGQA task format using LLMs to support the adaptation of PLM to the special input format and attention mechanism.

## 2 Related Work

**Question Answering over Knowledge Graph.** Multi-hop KGQA aims to find answer entities that are multiple hops away from the topic entities in a large-scale KG. Existing work (Sun et al., 2018) typically first retrieves a question-relevant subgraph from the KG to reduce the search space and then performs multi-hop reasoning to find the answer entities. Several methods (Sun et al., 2018; He et al., 2021) have been developed to facilitate the answer reasoning over the KG. These methods typically consist of a question encoder to represent the question, and a reason module to perform multi-hop reasoning over the KG using the question representation. Early work (Sun et al., 2018, 2019) uses a simple LSTM to encode the question and a GNN to model the reasoning over KG. However, a singular representation of the entire question creates confusion for the GNN regarding the specific relation that should be attended to at each step. To address this concern, subsequent work (He et al., 2021) attempts to decompose the semantics of the question and send the corresponding representation to the GNN module at each step. With the development of PLMs, recent work (Shi et al., 2021; Jiang et al., 2022b) proposes to enhance the question understanding by using PLM as the question encoder. They use PLM to compute the semantic similarity between the question and relations at each step and use a simpler GNN to propagate similarity information and update entity scores over KG.

**PLM for Knowledge Graph Reasoning.** Besides KGQA, PLM has also been used for other knowledge graph reasoning tasks, such as commonsense reasoning (Yasunaga et al., 2021) or predicting missing facts (Zamini et al., 2022). There are mainly two methods for PLM to leverage the KG. Several studies (Yasunaga et al., 2021) attempt to fuse the representation of the KG into the PLM, which is modeled by the GNN. It can make the PLM aware of the KG to some extent through the modeled representation. However, it's not easy to bridge the gap between the PLM and GNN given

the different model architecture and initialized parameters, leading to a sufficient understanding of KG for PLM. In contrast, a more direct way is to linearize the KG as a sequence and input it to PLMs (Xie et al., 2022; Saxena et al., 2022). In this way, the PLM can directly utilize the KG to perform reasoning. Despite its simplicity, such a way neglects the structure of KG, which is an important feature. In contrast, we propose to model the question and KG in a single PLM, while capturing the structure information with subgraph-aware self-attention mechanism.

## 3 Preliminary

In this section, we present the notations utilized throughout the paper, followed by a formal definition of the KGQA task.

**Knowledge Graph (KG).** A knowledge graph is commonly composed of a collection of triples, expressed as $\mathcal{G} = \{\langle e, r, e'\rangle | e, e' \in \mathcal{E}, r \in \mathcal{R}\}$, where $\mathcal{E}$ and $\mathcal{R}$ denote the entity set and relation set, respectively. A triple $\langle e, r, e'\rangle$ describes the fact that a relation $r$ exists between head entity $e$ and tail entity $e'$. Furthermore, we introduce *entity neighborhood* to denote both incoming and outgoing triples for an entity $e$, denoted as $\mathcal{N}_e = \{\langle e, r, e'\rangle \in \mathcal{G}\} \cup \{\langle e', r, e\rangle \in \mathcal{G}\}$. In this way, we can simplify the definition of the neighborhood triples for an entity $e$ as $\mathcal{N}_e = \{\langle e, r, e'\rangle \in \mathcal{G}\}$.

**Question Answering over Knowledge Graph (KGQA).** Given a natural language question $q$ and a KG $\mathcal{G}$, the task of KGQA aims to find answer entitie(s), denoted as $\mathcal{A}_q \in \mathcal{E}$, to the question on the KG. Following previous work (Sun et al., 2018), we assume that the entities mentioned in the question have already been linked with entities on KG, called *topic entities*, denoted as $\mathcal{T}_q \subset \mathcal{E}$. In this work, we focus on solving the *KGQA* task where the answer entities are multiple hops away from the topic entities over the KG. Considering the trade-off between efficiency and accuracy, we follow existing work (Sun et al., 2018) that solves this task using a *retrieval-then-reasoning* framework. In the two-stage framework, given a question $q$ and topic entities $\mathcal{T}_q$, the retrieval stage aims to retrieve a small question-relevant subgraph $\mathcal{G}_q$ from the large-scale input KG $\mathcal{G}$, while the reasoning stage finds answer entities $\mathcal{A}_q$ by reasoning over the retrieved subgraph $\mathcal{G}_q$.

## 4 Approach

In this section, we present our proposed approach, *i.e.,* ReasoningLM, which enables both effective question understanding and KG reasoning in a single PLM for better solving the KGQA task.

### 4.1 Overview

Existing KGQA methods (Sun et al., 2018) typically adopt a PLM to encode the question into latent representation, and a GNN to perform the reasoning over KG guided by the representation. However, due to the architecture divergence, it is hard to closely integrate the PLM and GNN for knowledge sharing and feature interaction. Inspired by recent work (Dwivedi and Bresson, 2020) that reveals the similarity between GNN and Transformer architecture of PLMs, we take a different perspective to simplify the above two-module approach and develop a more capable PLM that can support structural subgraph reasoning for KGQA.

Our basic idea is to implement a GNN within the PLM, to bridge the knowledge gap for question understanding and KG reasoning. To achieve it, we propose the ReasoningLM, enabling both effective question understanding and KG reasoning in a unified approach. Concretely, we first adapt the PLM to implement the functionality of GNN to aggregate the information from the neighboring entities and relations with *subgraph-aware self-attention* mechanism (Section 4.2). Then, we adopt *adaptation tuning* to optimize the PLM parameters to better adapt it to the special input format and attention mechanism (Section 4.3). Finally, we apply the adapted PLM to solve the KGQA task with parameter-efficient fine-tuning (Section 4.4).

The overview of ReasoningLM is shown in Figure 1. In our model, as the PLM can understand the question and perform reasoning in KG in a unified approach, it can freely share and interact with the knowledge from both for improving the KGQA task. Besides, such a way also enables the PLM to fully participate in the KG reasoning process (instead of GNN solely), which can make full use of its rich knowledge and strong reasoning capacity.

### 4.2 Adapting PLM for Subgraph Reasoning

In this section, we discuss how to adapt PLM for subgraph reasoning. Next, we introduce the subgraph data serialization and then present the subgraph-aware self-attention mechanism to unify question understanding and subgraph reasoning

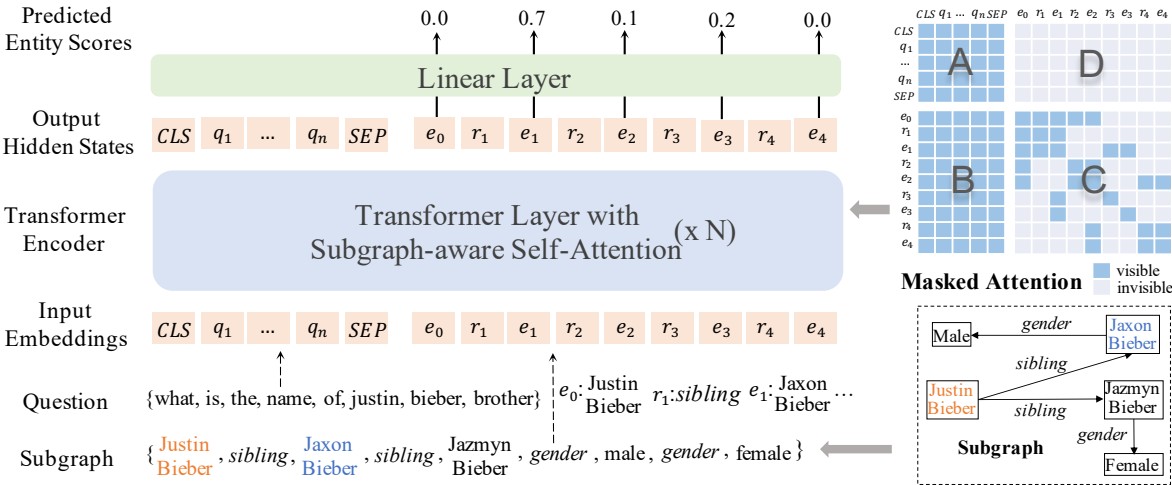

Figure 1: The illustration of performing answer entity reasoning over a subgraph according to the question using ReasoningLM with our proposed subgraph serialization and subgraph-aware self-attention.

within a single PLM, respectively.

### 4.2.1 BFS-based Subgraph Serialization

For KGQA task, the input data typically consists of the natural language question $q$ and its relevant subgraph $\mathcal{G}_q$ retrieved from the KG. To enable the PLM to capture the structured information from the subgraph and perform reasoning on it, we propose a breadth-first search (BFS) based subgraph serialization to convert the subgraph into a sequence.

Concretely, starting from a topic entity $e_0 \in \mathcal{T}_q$, we perform the BFS to visit all triples on $\mathcal{G}_q$. It first visits all the triples whose head entity is the topic entity, e.g., $\langle e_0, r_1, e_1 \rangle$, and then moves on to the triples whose head entities have been visited before, and so forth. Based on the order in the BFS process, we concatenate all the visited triples as a long sequence. To reduce the node sequence length, we only concatenate the entities or relations for the first time that they are visited, hence the final serialized subgraph consists of all the visited entities and relations, denoted as:

$$S_{\mathcal{G}_q} = \{e_0, r_1, e_1, r_2, e_2, ..., r_m, e_m\}, \quad (1)$$

where $m$ is the total number of triples in $S_{\mathcal{G}_q}$.

In this way, we can preserve the structure information of the subgraph within a relatively short sequence (with length as the count of entities and relations in $S_{\mathcal{G}_q}$), which can be varied according to the context length of language models.

### 4.2.2 Subgraph-aware Self-Attention

Based on the serialized subgraph, we leverage the subgraph-aware self-attention mechanism to support graph reasoning within the PLM, to propagate relevance information along with the relation edges on KG. We first initialize the embeddings of the serialized subgraph and the question, and then perform self-attention with constrained masks to aggregate their representations.

**Embeddings Mapping.** Since PLMs do not have the mapping embeddings for the entities and relations within the serialized subgraph, we need to initialize their embeddings before feeding them into the PLM. To embed them into the semantic space of the PLM, we first tokenize each entity or relation into subwords[1] using the tokenizer of the PLM, and then sum their embeddings into a single embedding to represent it. Finally, we can obtain the embedding matrix of the whole serialized subgraph, denoted as $\boldsymbol{N}_{\mathcal{G}_q} \in \mathbb{R}^{l \times d}$, where $d$ is the embedding dimension and $l$ is the length of the input sequence. Next, we concatenate it with the token embedding matrix $\boldsymbol{N}_q$ of the question $q$ after tokenization, to compose the input token embedding matrix of the PLM:

$$\boldsymbol{N} = [\boldsymbol{N}_q; \boldsymbol{N}_{\mathcal{G}_q}] = [\boldsymbol{n}_{q_1}, \cdots, \boldsymbol{n}_{q_n}; \boldsymbol{n}_{e_0}, \cdots, \boldsymbol{n}_{e_m}]$$

Based on it, we also add the position embeddings as $\boldsymbol{N}_E = \boldsymbol{N} + \boldsymbol{E}$ to obtain the input embedding matrix of the PLM.

**Self-Attention with Constrained Masks.** After obtaining the input embedding matrix $\boldsymbol{N}_E$, we feed it into the multi-layer Transformer encoder

---

[1]PLMs mostly use Byte-Pair Encoding tokenizer, which may segment an entity or relation name into several subwords.

of the PLM, to perform reasoning over the subgraph based on the given question. To enable the fine-grained semantic interaction between the question and the associated subgraph, we propose a constrained mask mechanism on the self-attention layers for controlling the attention interaction, including four kinds of attention modes:

• *Full question attention.* The representations of each token of question can attend to the other tokens of question (part A in Figure 1).

• *Full subgraph → question attention.* The representations of all the entities and relations can attend to the question representations, hence we can perform reasoning on the KG based on the question (part B in Figure 1).

• *Structural subgraph attention.* In the serialized subgraph, an entity can aggregate the information from its one-hop neighboring entities and relations (in a triple), similar to the updated way of node representations in GNN. Further, a relation can aggregate the information from its head and tail entities in the triple, as it establishes a link between the two entities (part C in Figure 1).

• Except for the above ways, other information flows are forbidden in the self-attention layer. In addition to the constraint on the subgraph structure, we also prevent the question attending to the subgraph, which can avoid the question representations to be influenced by the irrelevant information in the subgraph (part D in Figure 1).

To achieve them, we design the constrained self-attention mask $M \in \mathcal{R}^{l \times l}$, where the value in the $i$-th raw and $j$-th column denotes whether the $i$-th token can attend to the $j$-th one (0) or not (-inf), denoted as:

$$M_{ij} = \begin{cases} 0 & x_i \in S_{\mathcal{G}_q} \text{ and } x_j \in q, \\ 0 & x_i, x_j \in S_{\mathcal{G}_q} \text{ and } a_{ij} = 1, \quad (2) \\ \texttt{-INF} & \text{others}, \end{cases}$$

where $x_i$ and $x_j$ represent the tokens (*i.e.,* relations, entities or words) in the $i$-th and $j$-th positions of the input, and $a_{ij} = 1$ indicates that $x_i$ and $x_j$ are adjacent (within a KG triple). Then, we utilize the mask matrix to compute the constrained self-attention on the multi-layer Transformer encoder of the PLM as follows:

$$\text{Attn}(\boldsymbol{Q}, \boldsymbol{K}, \boldsymbol{V}) = \text{softmax}(\boldsymbol{A} + \boldsymbol{M})\boldsymbol{V}, \quad (3)$$

where $\boldsymbol{A} \in \mathcal{R}^{l \times l}$ is the original attention matrix, and $\boldsymbol{Q}, \boldsymbol{K}, \boldsymbol{V} \in \mathcal{R}^{l \times d}$ are the input representation

matrices of the self-attention layer. In this way, only the self-attention values between invisible tokens would be zero (after softmax activation on -INF values), avoiding the PLM to aggregate representations from them.

### 4.3 Adaptation Tuning

To help the PLM well adapt into the special input format and attention mechanism, we adopt the adaptation tuning to adapt the parameters of the PLM. We first collect the tuning dataset based on sampled subgraphs and synthesized questions, and then utilize the answer entity prediction task for training.

#### 4.3.1 Tuning Data Construction

To enable the PLM to understand the question and perform reasoning on the KG, we construct an adaptation dataset in an automatic way to tune its parameters. The dataset consists of 20,000 synthesized questions with relevant subgraphs extracted from the KG, and the answer entities. Next, we introduce the process of subgraph extraction and question synthesis.

**Subgraph Extraction.** We extract the subgraphs from Wikidata[2], a general-domain KG with natural language descriptions of the entities and relations. We consider to extract a set of subgraphs centering around popular entities, to better adapt the PLM for understanding commonly-used knowledge. Thus, we use the popular entities in Wikidata5M (Wang et al., 2021) as our seed topic entity set following KQA Pro (Cao et al., 2022), then randomly sample the answer and the subgraph. Starting from the topic entity, we perform a random walk over the KG, to sample a reasoning path with no more than 4 hops, whose ending entity is regarded as the answer entity. Then, we randomly extract the entities and relations around the topic entity to compose the subgraph, where we guarantee that the entities and relations from the reasoning path are also included. Such a way is easy to conduct and can automatically extract multiple subgraphs with the answer entities.

**Question Synthesis.** Based on the sampled reasoning path and answer entities, we also adopt an automatic way to synthesize the questions. Here, we propose two approaches, *i.e.,* rule-based synthesis and LLM-based synthesis. For the rule-based

---

[2]https://www.wikidata.org/

one, we first hand-craft several general templates, and then utilize them to convert the topic entity and the relations on the reasoning path into a natural language question. However, such a way leads to a poor diversity of questions, and also needs human efforts. Recently, as large language models (*e.g.,* ChatGPT) have shown a powerful generation capability (Brown et al., 2020; Zhao et al., 2023) and have been used to generate label-free datasets (Chen et al., 2023), we seek help from them to produce more high-quality questions. Concretely, we write a prompt to guide ChatGPT, a popular LLM, to generate the corresponding question for the answer entity based on the reasoning path and the topic entity. In this way, we cost approximately 15 dollars, and obtain 20,000 questions with diverse formats and fluent expression. The detail prompt we used is shown in Appendix D.

### 4.3.2   Answer Entity Prediction

Given the synthesized question, extracted subgraphs and the answer entities, we feed them into our ReasoningLM, and tune the model parameters via the answer entity prediction task. It is a multi-classification task to predict which entity in the subgraph is the answer entity of the question. Concretely, through the multi-layer Transformer encoder with constrained self-attention, we can obtain the hidden state $\boldsymbol{H} \in \mathcal{R}^{l \times d}$ of the input sequence at the last layer. Then, we add a linear prediction layer with the softmax activation function to transform the hidden state into the answer scores of all entities:

$$\boldsymbol{s} = \text{softmax}(\text{Linear}(\boldsymbol{H})), \qquad (4)$$

where $\boldsymbol{s} \in \mathcal{R}^l$. Then, we minimize the KL divergence between the predicted and ground-truth answer scores as:

$$\mathcal{L}_{at} = D_{KL}(\boldsymbol{s}, \boldsymbol{s}^\star), \qquad (5)$$

where $\boldsymbol{s}^\star$ is the ground-truth answer scores of all entities, where an entity is 1 if it is a labeled answer entity. Note that we only compute the loss for the entities, as the relations and question words are not able to be the answer.

### 4.4   Efficient Fine-tuning

After adaptation tuning, the PLM has been well adapted to performing reasoning over the general-domain subgraphs. Therefore, we further perform parameter-efficiently fine-tuning on the PLM, to apply it to the subgraph retrieval and answer reasoning subtasks respectively, where we only tune the parameters in the adapters (Houlsby et al., 2019) but freeze other parameters.

**Subgraph Retrieval.**   We follow Zhang et al. (2022) to fine-tune our model on the subgraph retrieval subtask, where we optimize our model to learn to predict the similarity between the question and relevant relations. During inference, starting from the topic entities, the model iteratively measures the semantic relevance between the question and neighboring relations, and adds proper ones and their corresponding triples into the subgraph, to extract a question-relevant subgraph.

**Answer Reasoning.**   Based on the retrieved subgraph, we also utilize the answer entity prediction task in Section 4.3.2 to fine-tune our ReasoningLM, to learn to accurately find the answer entities of the given question from the subgraph. During inference, we select the highest scoring entity predicted by our approach as the answer entity.

## 5   Experiments

### 5.1   Experimental Setup

**Adaptation Tuning Corpus**. Our adaptation tuning corpus is collected from Wikidata (Tanon et al., 2016), a general domain knowledge graph. We download the English Wikidata Dumps (2018/12/31) from the official site, and extract 2,000 entities from Wikidata5M (Wang et al., 2021) as seed topic entities. Finally, we construct 20,000 samples for adaptation tuning and split 1,000 samples as the validation set for selecting the best checkpoint.

**Datasets**. Following existing work on KGQA (He et al., 2021), we conduct experiments on three popular datasets to evaluate our proposed approach, including *WebQuestionsSP (WebQSP)* (Yih et al., 2015), *Complex WebQuestions 1.1 (CWQ)* (Talmor and Berant, 2018), and *MetaQA (MQA)* (Zhang et al., 2018). Table 5 shows the statistics of the three datasets. We give a detailed description of each dataset in Appendix A

**Evaluation Protocol**. We follow existing work that treats the reasoning as a ranking task for evaluation (Sun et al., 2018). For each question, we rank the answer score of all candidate entities and then assess the correctness of the top-1 answer using the

Table 1: Performance comparison of different methods for KGQA (Hits@1 and F1 in percent). We copy the results of LLMs from Jiang et al. (2023) and the results of the other baselines from Jiang et al. (2022b). Bold and underline fonts denote the best and the second-best methods, respectively. "FPT" and "PET" denote the full-parameter tuning and parameter-efficient tuning, respectively. "Rule-SYN" and "LLM-SYN" refer to synthesize the questions using rule-based and LLM-based strategies, respectively.

| Models | Updated Params | WebQSP | | CWQ | | MQA-1H | MQA-2H | MQA-3H |
| | | Hits@1 | F1 | Hits@1 | F1 | Hits@1 | Hits@1 | Hits@1 |
|---|---|---|---|---|---|---|---|---|
| KV-Mem (Miller et al., 2016a) | - | 46.7 | 34.5 | 18.4 | 15.7 | - | - | - |
| GraftNet (Sun et al., 2018) | 0.5M | 66.4 | 60.4 | 36.8 | 32.7 | 82.5 | - | - |
| PullNet (Sun et al., 2019) | - | 68.1 | - | 45.9 | - | - | - | - |
| EmbedKGQA (Saxena et al., 2020) | 125M | 66.6 | - | - | - | 92.0 | 40.7 | 34.6 |
| NSM (He et al., 2021) | 3M | 68.7 | 62.8 | 47.6 | 42.4 | 94.8 | 97.0 | 91.0 |
| TransferNet (Shi et al., 2021) | 111M | 71.4 | - | 48.6 | - | 96.5 | 97.5 | 90.1 |
| SR+NSM+E2E (Zhang et al., 2022) | 130M | 69.5 | 64.1 | 49.3 | 46.3 | - | - | - |
| UniKGQA (Jiang et al., 2022b) | 12M | 75.1 | 70.2 | 50.7 | 48.0 | **97.1** | 98.2 | 92.6 |
| Davinci-003 (Ouyang et al., 2022) | - | 48.3 | - | - | - | 52.1 | 25.3 | 42.5 |
| ChatGPT | - | 61.2 | - | - | - | 61.9 | 31.0 | 43.2 |
| StructGPT (Jiang et al., 2023) | - | 72.6 | - | - | - | 94.2 | 93.9 | 80.2 |
| ReasoningLM (FPT, LLM-SYN) | 1M | **78.5** | **71.0** | **69.0** | **64.0** | 96.5 | **98.3** | **92.7** |
| w FPT, Rule-SYN | 1M | 78.0 | 70.5 | 62.8 | 55.4 | 96.1 | 96.9 | 91.0 |
| w PET, LLM-SYN | 1M | 76.7 | 69.1 | 68.3 | 62.4 | 95.7 | 97.0 | 90.9 |

Table 2: Statistics of the experiment datasets.

| Type | Task | KG | Train | Dev | Test |
|---|---|---|---|---|---|
| | WebQSP | Freebase | 2,848 | 250 | 1,639 |
| | CWQ | Freebase | 27,639 | 3,519 | 3,531 |
| KGQA | MQA-1H | OMDb | 161 | 9,992 | 9,947 |
| | MQA-2H | OMDb | 210 | 14,872 | 14,872 |
| | MQA-3H | OMDb | 150 | 14,274 | 14,274 |

*Hits@1* metric. Given that a question may have multiple answers, we also adopt the *F1* metric.

**Baselines**. We consider the following three types of baseline methods for performance comparison: (1) *non PLM-based* methods: KV-Mem (Miller et al., 2016b), GraphtNet (Sun et al., 2018), PullNet (Sun et al., 2019), NSM (He et al., 2021); (2) *PLM-based* methods: Embed-KGQA (Saxena et al., 2020), TransferNet (Shi et al., 2021), SR+NSM+E2E (Zhang et al., 2022), UniKGQA (Jiang et al., 2022b); (3) *LLM-based* methods: Davinci-003 (Ouyang et al., 2022), Chat-GPT, StructGPT (Jiang et al., 2023). We give a detailed description of each baseline in Appendix B.

## 5.2 Implementation Details

In our experiment, we use RoBERTa-base as our base PLM. During adaptation tuning, we optimize parameters with the AdamW optimizer, where the batch size is 40 and the learning rate is 1e-4. We select the best checkpoint of adaptation tuning according to the evaluation of the constructed vali-

dation set. After adaptation tuning, we apply the ReasoningLM to downstream KGQA tasks with parameter-efficient fine-tuning. We add the extra retrieval and reasoning adapter to the ReasoningLM for subgraph retrieval and answering reasoning respectively, and only update the reasoning adapter while freezing other parameters.

For the retrieval stage, we follow the pipeline of Zhang et al. (2022) to fine-tune our ReasoningLM and then perform subgraph retrieval. we collect question-relation pairs based on the shortest relation paths between topic entities and answer entities, and then use these pairs to fine-tune the model to compute the similarity between the question and relations. We directly compute the similarity between the question and relations at each hop and freeze other parameters except for the adapter module. we optimize parameters with the AdamW optimizer, where the batch size is 10 and the learning rate is 5e-5 for all datasets. Then, we leverage the model to retrieve the subgraph. Specifically, starting from the topic entities, the model iteratively measures the semantic relevance between the question and neighboring relations, and adds top-$k$ relations and their corresponding triples into the subgraph. We set the $k$ as 15 for WebQSP and CWQ, and 3 for MetaQA.

For the reasoning stage, we fine-tune ReasoningLM on the retrieved subgraph to perform answer reasoning. Similarly, we only update the adapter model and freeze other parameters. We optimize

parameters with the AdamW optimizer, where the batch size is 4 for MetaQA, 60 for WebQSP, and 300 for CWQ while the learning rate is 1e-4 for all datasets.

## 5.3 Main Results

We show the results of all five data in Table 1. First, directly using LLMs (*e.g.,* Davinci-003 and ChatGPT) can achieve performance comparable to part of the supervised learning baselines on the WebQSP dataset. However, LLMs perform not well on the more complex multi-hop datasets, such as MQA-2H and MQA-3H. It demonstrates that LLMs are still hard for effectively solving KGQA tasks relying solely on the LLM. Despite the incorporation of external KG can enhance LLMs (*i.e.,* StructGPT), they still have a performance gap compared to the strong supervised learning models.

Secondly, PLM-based methods (*e.g.,* Transfer-Net, SR+NSM+E2E, and UnikGQA) can achieve a consistently better performance compared with methods not using PLM (*e.g.,* GraftNet, PullNet, and NSM).And UniKGQA achieves a further performance improvement on all datasets, benefiting from its unified architecture to learn the essential capability of question-relation semantic matching for both retrieval and reasoning stages.

Finally, our ReasoningLM is substantially better than all other competitive baselines in all datasets with only updating a few parameters (only 1M), achieving a 4.5% improvement of Hits@1 on WebQSP and 36.1% improvement of Hits@1 on more difficult CWQ compared to the best baseline. Unlike the other baselines, our approach develops a subgraph reasoning enhanced PLM to model the question and subgraph seamlessly. We can utilize the pre-trained knowledge within PLM, while enabling the reasoning process over the subgraph can directly attend to the question. With further adaptation tuning, our model can be applied to downstream tasks with parameter-efficient fine-tuning. These results demonstrate the effectiveness and efficiency of our ReasoningLM model.

In our approach, we update the full parameters of our model (FPT) during adaptation tuning with LLM synthesis data (LLM-SYN). Actually, we can accomplish adaptation tuning at a smaller cost by updating fewer parameters or using cheaper constructed data. Here, we study it by proposing two variants of our ReasoningLM: (1) *w FPT, Rule-SYN* that updates the full pa-

Table 3: Ablation study of our the subgraph-aware self-attention mechanism (SA) and adaptation tuning (AT).

| Models | WebQSP | | CWQ | |
|---|---|---|---|---|
| | Hits@1 | F1 | Hits@1 | F1 |
| ReasoningLM | 78.5 | 70.1 | 69.0 | 64.0 |
| *w/o* SA | 68.5 | 63.2 | 40.5 | 38.2 |
| *w/o* AT | 67.5 | 60.4 | 55.2 | 43.3 |

Table 4: Performance of implementing ReasoningLM with different PLMs on CWQ.

| CWQ | RoBERTa (base) | RoBERTa (large) | DeBERTa (base) | BERT (base) |
|---|---|---|---|---|
| Hits@1 | 69.0 | 70.0 | 68.1 | 67.4 |
| F1 | 64.0 | 65.2 | 63.1 | 63.0 |

rameters with rule-based constructed data, (2) *w PET, LLM-SYN* that updates an added adapter while freezing the model parameters with LLM synthesis data. We can see that although both variants show varying degrees of performance decline, they can still achieve better results compared to the existing baselines on WebQSP and CWQ. For MetaQA, the two variants only achieve comparable performance to existing baselines. A possible reason is that limited data makes it difficult to adapt the model to a specific domain.

## 5.4 Further Analysis

**Ablation Study.** Our approach contains two important technical contributions, the first is the subgraph-aware self-attention mechanism (SA), which imitates the GNN reasoning over the graph, and the second is the adaptation tuning (AT), which enhances the reasoning capability over KG. Here, we conduct the ablation study to verify their effectiveness by removing each one individually. We experiment with two variants as: (1) *w/o AT* removing the adaptation tuning procedure, (2) *w/o SA* removing the subgraph-aware self-attention mechanism in self-attention mechnism. We show the results of the ablation study in Table 3. All these variants underperform the complete ReasoningLM, which indicates that the two strategies are both important for developing a subgraph reasoning enhanced PLM for KGQA. Besides, we also analyze combining other reasoning models (*e.g.,* NSM and UniKGQA) with our retrieval sugraphs in Appendix C.

**Variants with Different PLMs.** Since our proposed adaptation method does not change the orig-

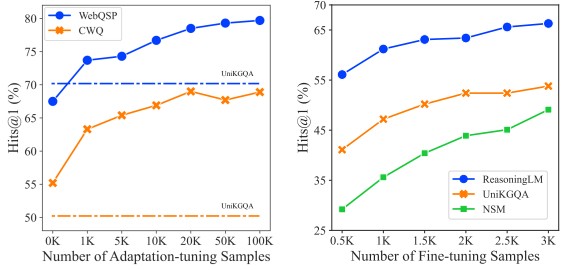

Figure 2: The Hits@1 scores of our ReasoningLM on WebQSP and CWQ after adaptation tuning with a various number of samples (Left). And the Hits@1 score of our ReasoningLM compared with two strong baselines (*i.e.,* NSM and UniKGQA) on CWQ when fine-tuning with various numbers of samples (Right)

inal model architecture, it can be applied to other different PLMs. To explore the performance with different PLMs, we conduct experiments with other three PLMs (*i.e.,* RoBERTa-large, DeBERTa, and BERT). We show the results in Table 4. It is observed that the utilization of DeBERTa-base and BERT-base also yields performance comparable to RoBERTa-base. This suggests that our adaptation method is agnostic to the PLMs used. At the same time, a larger RoBERTa-large can achieve further performance improvement compared with RoBERTa-base. It indicates the potential of our method to be applied to larger PLMs. Limited by computational resources, we would apply our method to larger PLMs in future work.

**Adaptation Tuning Efficiency.** Although the adaptation tuning strategy is important to enhance the reasoning capability of ReasoningLM, too many tuning samples require significant construction and tuning costs. Here, we investigate the downstream performance of ReasoningLM on WebQSP and CWQ *w.r.t.* varying numbers of adaptation tuning samples. As shown in Figure 2, we can see that the ReasoningLM can reach a competitive performance compared with the best baseline UniKGQA after adaptation tuning with a few samples (*i.e.,* 5K). It shows that our approach does not require too much data to complete a successful adaptation tuning. Simultaneously, we can observe that as the number of tuning data increases, our model's performance will improve even further and eventually reach a stable state. It indicates that we only need a few tuning examples to achieve a trade-off between the tuning costs and downstream performance.

**Fine-tuning Efficiency.** As our ReasoningLM

model has become familiar with the multi-hop reasoning over the subgraph after adaptation tuning, it can be easily applied to specific downstream KGQA tasks with fewer labeled samples, which is meaningful for sparse data scenarios. To explore it, we compare the final performance changes of our ReasoningLM with two strong baselines UniKGQA and NSM *w.r.t.* the increasing of fine-tuning samples with the same retrieval model. We conduct experiments using CWQ, which is more challenging and has a larger training set. The results are presented on the right of Figure 2. ReasoningLM can obtain consistent performance improvements compared with other two baselines under various numbers of tine-tuning samples. It indicates that our ReasoningLM has a better understanding of the answer reasoning over the KG.

## 6 Conclusion

In this work, we proposed a subgraph reasoning enhanced PLM to support question understanding and KG reasoning in a single PLM, namely ReasoningLM. In our approach, we first adopted a BFS-based subgraph serialization to enable the PLM to capture the structured information and then proposed a subgraph-aware self-attention mechanism to support graph reasoning within the PLM based on the serialized subgraph. In order to adapt the PLM to the special input format and attention mechanism, we further utilized an adaptation tuning strategy with a cheap data construction cost of 15 dollars by using ChatGPT in an automatic way. Finally, we applied the ReasoningLM to solve downstream KGQA tasks with parameter-efficient fine-tuning. Experimental results have shown that our approach can significantly improve the performance compared to existing strong baselines by updating only 1M parameters.

## 7 Limitations

In our approach, we propose a subgraph reasoning enhanced PLM by adapting existing PLM without modifying its original architecture. Therefore, the input is usually limited (*e.g.,* 512) for most of PLMs, causing our model unable to process the arbitrary size of the retrieved subgraph. We can relieve it by using the relative position embedding or a better retrieval model to obtain the proper size of the subgraph. In addition, although we conduct experiments on multiple KGQA datasets, there is a lack of evaluation on other KG reasoning

tasks, such as commonsense question answering and knowledge graph completion. They can be transformed into the same input format, which can be potentially solved with our method. From Table 4, it can be seen that increasing the model parameters will lead to further performance improvement. However, due to limited computational resources, we did not conduct experiments on larger PLMs (*e.g.,* more than 1 Billion parameters).

## Acknowledgments

This work was partially supported by Beijing Natural Science Foundation under Grant No. 4222027, National Natural Science Foundation of China under Grant No. 62222215, and the Outstanding Innovative Talents Cultivation Funded Programs 2022 of Renmin University of China. This work was also supported by Alibaba Group through Alibaba Innovative Research Program. Xin Zhao is the corresponding author.

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

## A Datasets

Following existing work on KGQA (He et al., 2021), we conduct experiments on three popular datasets to evaluate our proposed approach, including *WebQuestionsSP (WebQSP)* (Yih et al., 2015), *Complex WebQuestions 1.1 (CWQ)* (Talmor and Berant, 2018), and *MetaQA (MQA)* (Zhang et al., 2018).

• **MetaQA** (Zhang et al., 2018) comprises over 400,000 questions in the movie domain, with answer entities located up to three hops away from the topic entities. Based on the number of hops, the dataset is divided into three sub-datasets: MetaQA-1hop, MetaQA-2hop, and MetaQA-3hop. Existing work has demonstrated that the training data for MetaQA is *more than sufficient* (He et al., 2021), hence all the comparison methods in our experiments can achieve very high performance. We randomly sample just one training case for each question template from the original training set, to form a one-shot training dataset following existing work (He et al., 2021; Jiang et al., 2022b). In this way, the numbers of training samples for MetaQA-1hop, MetaQA-2hop, and MetaQA-3hop are 161, 210, and 150, respectively.

• **WebQuestionsSP (WebQSP)** (Yih et al., 2015) consists of 4,737 questions. The answer entities are within a maximum of 2 hops from the topic entity on the Freebase (Bollacker et al., 2008) KG. We adopt the train/valid/test splits from GraftNet (Sun et al., 2018) for consistency.

• **Complex WebQuestions 1.1 (CWQ)** (Talmor and Berant, 2018) is constructed based on WebQSP, which is more challenging. It complicates WebQSP by extending the question entities or adding constraints to restrict the answers. The answer entities are within a maximum of 4 hops from the topic entity on the Freebase (Bollacker et al., 2008) KG.

Existing work (He et al., 2021) has demonstrated that the training data for MetaQA is *more than sufficient*. To better reflect the reasoning capability of baselines and our method, we extract one sample for each question template and conduct the one-shot experiment on all three MetaQA sub-datasets following existing work (He et al., 2021). Table 5 shows the statistics of the three datasets.

## B Baselines

We consider the following three types of baseline methods for performance comparison: (1) *non PLM-based* methods: KV-Mem (Miller et al.,

Table 5: Statistics of the experiment datasets.

| Type | Task | KG | Train | Dev | Test |
|------|------|-----|-------|-----|------|
| KGQA | WebQSP | Freebase | 2,848 | 250 | 1,639 |
| | CWQ | Freebase | 27,639 | 3,519 | 3,531 |
| | MQA-1H | OMDb | 161 | 9,992 | 9,947 |
| | MQA-2H | OMDb | 210 | 14,872 | 14,872 |
| | MQA-3H | OMDb | 150 | 14,274 | 14,274 |

2016b), GraphtNet (Sun et al., 2018), Pull-Net (Sun et al., 2019), NSM (He et al., 2021); (2) *PLM-based* methods: EmbedKGQA (Saxena et al., 2020), TransferNet (Shi et al., 2021), SR+NSM+E2E (Zhang et al., 2022), UniKGQA (Jiang et al., 2022b); (3) *LLM-based* methods: Davinci-003 (Ouyang et al., 2022), ChatGPT, StructGPT (Jiang et al., 2023). We give a detailed description of each baseline:

• **KV-Mem** (Miller et al., 2016b) employs a key-value memory table to store KG facts and facilitates multi-hop reasoning through iterative read operations on the memory.

• **GraftNet** (Sun et al., 2018) first utilize a heuristic method to retrieve the question-relevant subgraph and text sentences from the Knowledge Graph (KG) and Wikipedia, respectively. Subsequently, it employs a graph neural network to conduct multi-hop reasoning on a heterogeneous graph constructed from the subgraph and text sentences.

• **PullNet** (Sun et al., 2019) trains a graph retrieval model instead of the heuristic way in GraftNet for the retrieval task, and then conducts multi-hop reasoning with GraftNet.

• **NSM** (He et al., 2021) first conducts retrieval following GraftNet and then adapts the neural state machine (Hudson and Manning, 2019) used in visual reasoning for multi-hop reasoning on the KG. It consists of a question understanding module based on LSTM and a graph reasoning module with the adapted neural state machine, which is a graph neural network in essence.

• **EmbedKGQA** (Saxena et al., 2020) transforms the multi-hop reasoning process of GraftNet into a link prediction task. This is achieved by comparing pre-trained entity embeddings with question representations derived from a Pre-trained Language Model (PLM).

• **TransferNet** (Shi et al., 2021) first conducts retrieval following GraftNet and then performs the multi-hop reasoning on a KG or a text-formed relation graph in a transparent framework. It consists of a PLM for question encoding and a graph neural

network for updating the relevance scores between entities and the question.

• **SR+NSM+E2E** (Zhang et al., 2022) first learns a PLM-based relation path retriever to conduct effective retrieval and then leverages NSM reasoner to perform multi-hop reasoning. The whole process is optimized in an end-to-end way.

• **UniKGQA** (Jiang et al., 2022b) is a unified model architecture based on PLMs for both retrieval and reasoning stages. It consists of PLM for computing the semantic similarity between the question and relation and a simple graph neural network for propagating the matching information.

• **Davinci-003** (Ouyang et al., 2022) and **ChatGPT** are both large language models developed by OpenAI. We can use their provided APIs to access them and solve KGQA tasks.

• **StructGPT** (Jiang et al., 2023) is a general framework for improving the zero-shot reasoning ability of LLMs over structured data, such as Knowledge Graph. It use an invoking-linearization-generation procedure that leverages LLMs to read and perform reasoning based on the interface of structured data.

## C  Ablation Study of Retrieval Subgraphs

We conduct experiments on two strong baselines (NSM and UniKGQA) with our retrieval subgraphs to explore the effect of the retrieval stage. We show the results in Table 6. We can see that the two baselines achieve consistent performance improvement with our retrieved subgraphs (63.43% Hits@1 of UniKGQA w Ours v.s. 50.7% Hits@1 of UniKGQA and 61.9% Hits@1 of NSM w Ours v.s. 47.6% Hits@1 of NSM). It indicates that our model can achieve a better retrieval compared to existing retrieval methods. Although enhanced with our retrieval methods, the performance of the two baselines still have a great gap with our ReasoningLM. This demonstrates the effectiveness of our ReasoningLM to perform multi-hop reasoning over the subgraph.

## D  Prompt for ChatGPT

Inspired by existing work (Taori et al., 2023), we show the prompt of generating questions used by ChatGPT in Table 7.

Table 6: Performance of ReasoningLM and two strong baselines (*i.e.*, NSM and UniKGQA) on CWQ based on our retrieval subgraphs represented by "w Ours".

| CWQ | ReasoningLM | NSM | NSM w Ours | UniKGQA | UniKGQA w Ours |
|---|---|---|---|---|---|
| Hits@1 | 69.0 | 47.6 | 61.9 | 50.7 | 63.43 |
| F1 | 64.0 | 42.4 | 50.1 | 48.0 | 57.65 |

---

Here are the guidelines for formulating a question based on the given factual triples, object of the question, and the answer:

1. A factual triple consists of a head entity, a tail entity, and their relationship, representing a real-world fact. For example, (Gino Finizio, sex or gender, male) indicates that Gino Finizio is male.

2. Your question should pertain to the provided entity based on the factual background, and the answer provided should align with the provided answer.

3. The question should include all of the given information as constraints, except for the provided answer, to ensure that all provided information is fully considered in deriving the answer.

4. Utilize as many different entities and relations as possible in the question to promote variety.

5. Questions should generally be one to two sentences and require no added content aside from the question itself.

6. Use subordinate clauses to link multiple triples in the question, excluding intervening entities when possible. For example, the knowledge "(Elevator Action, platform, Commodore 64), (Commodore 64, Giphy username, commodore)" can be conveyed as "What is the Giphy username for the platform of Elevator Action?"

We provide a set of 4 examples that you can reference:

Example 1:
Given the factual background: (Euler-Lagrange equation, discoverer or inventor, Leonhard Euler), (Leonhard Euler, student, Mikhail Golovin). Please generate a question about the "Euler-Lagrange equation" and the answer to the question should be "Mikhail Golovin".
The question is: Who is the student that coined the Euler-Lagrange equation?

Example 2:
Given the factual background: (Spokane, population, 208,916), (Spokane, point in time, 2007). Please generate a question about "Spokane" and the answer to the question should be "208,916".
The question is: In 2007, what is the population of Spokane?

Example 3:
Given the factual background: (Kristen Stewart, place of birth, Los Angeles), (Los Angeles, capital of, Los Angeles County). Please generate a question about "Kristen Stewart" and the answer to the question should be "Los Angeles".
The question is: What is the birth city of Kristen Stewart, which has the county seat of Los Angeles County?

Example 4:
Given the factual background: (Lampedusa Airport, country, Italy), (Italy, capital, Rome), (Italy, start time, 2 June 1946). Please generate a question about "Lampedusa Airport" and the answer to the question should be "Rome".
The question is: Starting in 1946, what was the capital of the country to which Lampedusa Airport belonged?

---

Table 7: Instruction of generating questions in our adaptation-tuning strategy.