# OpenReview forum: "ReasoningLM: Enabling Structural Subgraph Reasoning  in Pre-trained Language Models for Question Answering over Knowledge Graph"
_EMNLP/2023/Conference — EMNLP 2023 Main_

### Official Review · Reviewer_3YnV · 2023-08-05

**Soundness:** 4

**Excitement:**

4: Strong: This paper deepens the understanding of some phenomenon or lowers the barriers to an existing research direction.

**Missing References:**

[1] Ainslie, Joshua, et al. "ETC: Encoding long and structured inputs in transformers." arXiv preprint arXiv:2004.08483 (2020).

**Paper Topic And Main Contributions:**

The paper presents an approach to enhance subgraph reasoning in Knowledge Graph Question Answering (KGQA) by developing a more capable Pre-trained Language Model (PLM) that incorporates subgraph-aware self-attention and an adaptation tuning strategy. The proposed model, named ReasoningLM, outperforms state-of-the-art models in KGQA tasks. The authors conduct extensive experiments, demonstrating the effectiveness of their approach in handling structured reasoning tasks and adapting model parameters with synthesized subgraphs.

**Questions For The Authors:**

1. Could the authors provide performance metrics for the ablated system with PET (Partial Rule-SYN) to allow for a fair comparison with PET (LLM-SYN)? This would help determine which of the two approaches is more beneficial when provided the same amount of training data.

2. In Line 323, it would be helpful to elaborate on whether the position embeddings E respect the structural positions of subgraph elements. This clarification would provide insight into the model's understanding of subgraph structure.

3. The rationale behind the choice of using KL divergence loss instead of conventional cross-entropy loss (Eq. 5) is not explicitly discussed. Could the authors elaborate on the advantages of KL divergence in this context and whether it leads to substantial improvements?

4. Addressing the concerns about the BFS-based subgraph serialization is crucial. BFS-based subgraph serialization might incorporate irrelevant triples since not all triples in the k-hop subgraph are useful. The num of triples included can be very large when the k becomes large. How does the proposed approach handle cases where multiple potentially disjoint subgraphs are far apart? Additionally, clarifying the construction of subgraphs in such scenarios would enhance the understanding of the model's capabilities.

5. The paper mentions that the proposed approach requires knowledge of the Golden topic entity. Could the authors clarify whether this is a unique requirement of their approach or if it applies to other baseline methods as well?

**Reasons To Accept:**

1. **Well-Motivated Approach:** The proposed approach is well-motivated and addresses a significant challenge in KGQA, namely enhancing subgraph reasoning. Comprehensive experiments have been conducted to verify the effectiveness of the proposed approach.

2. **Broad Applicability:** The paper highlights the agnostic nature of the proposed approach with respect to PLMs, which is demonstrated through experiments in Table 3. This feature increases the versatility of the model and its potential applicability to different settings.

**Reasons To Reject:**

1. **Incremental Novelty:** The novelty of the proposed approach is somewhat incremental, as similar concepts of structural self-attention have been explored in other graph transformer variants, such as the ETC paper [1]. Additionally, the adaptation tuning strategy appears straightforward. To strengthen the paper's contribution, the authors could discuss more explicitly how their approach differs or improves upon these prior works.

2. **Fair Comparison:** The comparison with baseline methods raises concerns about fairness due to the availability of additional training data during the adaptation tuning stage (which has proven to be very crucial to the success of the approach). To address this, the authors should consider training the baseline models with the same additional data and conducting a fair comparison to better assess the effectiveness of the proposed approach.

[1] Ainslie, Joshua, et al. "ETC: Encoding long and structured inputs in transformers." arXiv preprint arXiv:2004.08483 (2020).

**Reproducibility:**

4: Could mostly reproduce the results, but there may be some variation because of sample variance or minor variations in their interpretation of the protocol or method.

**Reviewer Confidence:**

3: Pretty sure, but there's a chance I missed something. Although I have a good feel for this area in general, I did not carefully check the paper's details, e.g., the math, experimental design, or novelty.

---

> ### Author Rebuttal · Authors · 2023-08-29
>
> Thanks for your insightful suggestions and we have listed our response to your concerns as follows.
>
> **1.Incremental Novelty: The novelty of the proposed approach is somewhat incremental, as similar concepts of structural self-attention have been explored in other graph transformer variants, such as the ETC paper [1]. Additionally, the adaptation tuning strategy appears straightforward. To strengthen the paper's contribution, the authors could discuss more explicitly how their approach differs or improves upon these prior works.**
>
> Thanks for your suggestion and we will add this discussion and cite the relevant ETC paper in our paper. Existing graph transformer variants, such as the ETC model, mainly aim to model the structure information or scale up the input length by adjusting the transformer architecture. While our goal is to provide a new view for modeling the interaction and understanding between text question and graph structured KG, especially for building transferable and powerful KGQA model with efficient training cost. Concretely, we list the following three differences.
>
> Firstly, inspired by these graph transformer variants, we enable the PLM to simultaneously model question understanding, deep interaction between question and subgraph, and reasoning over subgraph by leveraging adapted subgraph-aware self-attention mechanism.
>
> Secondly, since we do not modify the original model architecture, our approach allows the PLM to utilize pre-trained knowledge to better understand textual relations and entities in the subgraph and question when solving KGQA tasks.
>
> Thirdly, since the PLM is originally trained by learning from general-purpose natural language text, we further propose an adaptation tuning strategy that utilizes a few synthesized data to adapt the PLM to the special input format and attention mechanism. We propose an automatic data construction method for the KGQA task format using LLMs, such as ChatGPT, which has not been explored previously. Our data construction method could inspire the future work about human-free KGQA training data construction with the help of LLMs, since human annotated KGQA datasets are expensive [1].
>
> In addition to these differences, our proposed method also boost the rethinking about the potential of PLM on KGQA tasks even other tasks that require understanding text information and graph structure information simultaneously, such as commonsense question answering [2] or drug modeling with textual description [3].
>
> **2.Fair Comparison: The comparison with baseline methods raises concerns about fairness due to the availability of additional training data during the adaptation tuning stage (which has proven to be very crucial to the success of the approach). To address this, the authors should consider training the baseline models with the same additional data and conducting a fair comparison to better assess the effectiveness of the proposed approach.**
>
> In fact, the adaptation tuning dataset is established using the Wikidata KG, which differs from the Freebase KG used in the downstream WebQSP and CWQ datasets, as well as the movie domain KG used in MetaQA dataset. Since most of the baselines model the subgraph reasoning with GNN, they require specific entity embeddings and relation embeddings for each dataset. Despite training the baseline models with the same additional data, it is still difficult for previous methods to utilize the learned knowledge during the adaptation tuning phrase when fine-tuned on the downstream tasks, since new entity embeddings and relation embeddings would be reinitialized for different KGs. Here, we conduct an extra ablation experiment to compare our model with some strong baseline models that are also trained using the same adaptation tuning data. The results indicate that the additional adaptation tuning data does not yield any improvement for the baseline models.
> \\begin{array} {|l|c|c|}
> \\hline
> & \\textrm{WebQSP (Hits@1)} & \\textrm{CWQ (Hits@1)} \\\\
> \\hline
> \\textrm{NSM} & 68.7 & 47.6 \\\\
> \\hline
>   \\textrm{UniKGQA} & 75.1 & 50.7 \\\\
> \\hline
>   \\textrm{NSM w adaptation tuning}  & 67.6 & 48.1 \\\\
> \\hline
>   \\textrm{UniKGQA w adaptation tuning}  & 74.9 & 51.1 \\\\
> \\hline
>   \\textrm{ReasoningLM w adaptation tuning (Ours)}  & 78.5 & 69.0 \\\\
> \\hline
> \\end{array}
>
> **3.Could the authors provide performance metrics for the ablated system with PET (Partial Rule-SYN) to allow for a fair comparison with PET (LLM-SYN)? This would help determine which of the two approaches is more beneficial when provided the same amount of training data.**
> Thanks for your suggestion! We show the performance of PET (Rule-SYN) in the following table. In the table, "FPT" and "PET" denote adaption tuning with full-parameter tuning and parameter-efficient tuning, respectively. "Rule-SYN" and "LLM-SYN" refer to synthesized tuning data using rule-based and LLM-based strategies, respectively. Our results demonstrate that high-quality LLM synthesized data is more beneficial than rule-based synthesized data in both full-parameter and parameter-efficient tuning settings, especially for the complex KGQA dataset like CWQ. Besides, using the same synthesized data, full-parameter tuning is obviously better than parameter-efficient tuning.
>
> \\begin{array} {|l|c|c|}
> \\hline
> & \\textrm{WebQSP (Hits@1)} & \\textrm{CWQ (Hits@1)} \\\\
> \\hline
> \\textrm{FPT + LLM-SYN} & 78.5 & 69.0 \\\\
> \\hline
> \\textrm{FPT + Rule-SYN} &  78.0 & 62.8 \\\\
> \\hline
> \\textrm{PET + LLM-SYN} &  76.7 & 68.3 \\\\
> \\hline
> \\textrm{PET + Rule-SYN} &  75.1 & 61.3 \\\\
> \\hline
> \\end{array}
>
> **4.In Line 323, it would be helpful to elaborate on whether the position embeddings E respect the structural positions of subgraph elements. This clarification would provide insight into the model's understanding of subgraph structure.**
>
> In our implementation, we do not modify the original position embedding mechanism and do not utilize the position embeddings to represent the structural position, as our prior experiments have shown that it does not bring performance improvements and even hurt the model performance. The possible reason is that it affects the original natural language understandings of PLM, such new knowledge is hard to be efficiently learned by our few augmented data.  Interestingly, we find that our attention distribution can well match the subgraph structure to some extent.
>
> **5.The rationale behind the choice of using KL divergence loss instead of conventional cross-entropy loss (Eq. 5) is not explicitly discussed. Could the authors elaborate on the advantages of KL divergence in this context and whether it leads to substantial improvements?**
>
> The model produces a probability distribution over the subgraph's entities to indicate potential answers for the question. Additionally, the label is a probability distribution of each entity in the subgraph where the answer entity's position is assigned 1 and the remaining entities as 0. For multiple answers, we normalize the distribution. To minimize the distance between these two probability distributions, we utilize the KL divergence loss, as commonly used in other baselines. Alternatively, the cross-entropy loss can be used, which has the same effect as the objective function but differs by a constant mathematically. We select the KL divergence loss in this study for convenient comparisons with existing works.
>
> **6.Addressing the concerns about the BFS-based subgraph serialization is crucial. BFS-based subgraph serialization might incorporate irrelevant triples since not all triples in the k-hop subgraph are useful. The num of triples included can be very large when the k becomes large. How does the proposed approach handle cases where multiple potentially disjoint subgraphs are far apart? Additionally, clarifying the construction of subgraphs in such scenarios would enhance the understanding of the model's capabilities.**
>
> In fact, the scale of the subgraph is relatively small, since we have first performed a subgraph retrieval from the original KG following the mainstream retrieval-then-reasoning paradigm used in existing work [1-4]. The retrieval stage focuses on extracting a relatively small subgraph involving the answer entities. The reasoning stage aims to accurately find the answer entities of the given question by walking along the relations starting from the topic entities. Typically, the number of triples in a subgraph is less than 500. We only retrieve the subgraph starting from one topic entity and avoid cases where multiple disjoint subgraphs correspond to distant topic entities. Concretely, starting from the topic entity, our model, fine-tuned on the subgraph retrieval subtask following existing work [2][5] , iteratively measures the semantic relevance between the question and neighbouring relations, and adds proper ones and their corresponding triples into the subgraph. In this way, a smaller but more question-relevant subgraph would be constructed.
>
> **7.The paper mentions that the proposed approach requires knowledge of the Golden topic entity. Could the authors clarify whether this is a unique requirement of their approach or if it applies to other baseline methods as well?**
>
> For the adaptation tuning stage, since the tuning data is constructed in an automatic way, we can directly obtain the topic entity and use it when performing adaptation tuning. After adapatation tuning, we do not assume the given golden topic entity when fine-tuned on downstream KGQA tasks. And we just follow existing work [4-8] to perform entity linking in advance with available tools (e.g., Google Knowledge Graph Search API) and models (e.g., ELQ [9]). Our experiment settings are consistent with other baselines, which  makes sure a fair comparison in Table 2.
>
> Reference:
>
> [1] Yu Gu, Sue Kase, Michelle Vanni, Brian M. Sadler, Percy Liang, Xifeng Yan, Yu Su: Beyond I.I.D.: Three Levels of Generalization for Question Answering on Knowledge Bases. WWW 2021.
>
> [2] Michihiro Yasunaga, Hongyu Ren, Antoine Bosselut, Percy Liang, Jure Leskovec: QA-GNN: Reasoning with Language Models and Knowledge Graphs for Question Answering. NAACL 2021.
>
> [3] Philipp Seidl, Andreu Vall, Sepp Hochreiter, Günter Klambauer: Enhancing Activity Prediction Models in Drug Discovery with the Ability to Understand Human Language. ICML 2023.
>
> [4] Haitian Sun, Bhuwan Dhingra, Manzil Zaheer, Kathryn Mazaitis, Ruslan Salakhutdinov, William W. Cohen: Open Domain Question Answering Using Early Fusion of Knowledge Bases and Text. EMNLP 2018.
>
> [5] Apoorv Saxena, Aditay Tripathi, Partha P. Talukdar: Improving Multi-hop Question Answering over Knowledge Graphs using Knowledge Base Embeddings. ACL 2020.
>
> [6] Gaole He, Yunshi Lan, Jing Jiang, Wayne Xin Zhao, Ji-Rong Wen: Improving Multi-hop Knowledge Base Question Answering by Learning Intermediate Supervision Signals. WSDM 2021.
>
> [7] Jiaxin Shi, Shulin Cao, Lei Hou, Juanzi Li, Hanwang Zhang: TransferNet: An Effective and Transparent Framework for Multi-hop Question Answering over Relation Graph. EMNLP 2021.
>
> [8] Jinhao Jiang, Kun Zhou, Xin Zhao, Ji-Rong Wen: UniKGQA: Unified Retrieval and Reasoning for Solving Multi-hop Question Answering Over Knowledge Graph. ICLR 2023.
>
> [9] Belinda Z. Li, Sewon Min, Srinivasan Iyer, Yashar Mehdad, Wen-tau Yih: Efficient One-Pass End-to-End Entity Linking for Questions. EMNLP 2020.

---

### Official Review · Reviewer_UNw1 · 2023-08-05

**Soundness:** 3

**Excitement:**

4: Strong: This paper deepens the understanding of some phenomenon or lowers the barriers to an existing research direction.

**Paper Topic And Main Contributions:**

This paper focus on the knowledge graph question answering problem. The authors argue that existing methods utilize LM to model queries and utilize GNNs to model KGs. However, they are not combined thoroughly. This paper proposes a method to fuse the above methods and proposes ReasoningLM. The experiments on several public datasets demonstrate the effectiveness of this approch.

**Questions For The Authors:**

None

**Reasons To Accept:**

1.This paper is well-written and easy to follow.
2.The experimental results prove the effectiveness of the proposed method.
3.The method is simple and straightforward.

**Reasons To Reject:**

1.The method is incremental compared to existing methods and the self-attention mechanism is also widely used in other KBQA studies.
2.More case studies and error case studies should be performed to help see the benefit of the proposed method.

**Reproducibility:**

3: Could reproduce the results with some difficulty. The settings of parameters are underspecified or subjectively determined; the training/evaluation data are not widely available.

**Reviewer Confidence:**

3: Pretty sure, but there's a chance I missed something. Although I have a good feel for this area in general, I did not carefully check the paper's details, e.g., the math, experimental design, or novelty.

---

> ### Author Rebuttal · Authors · 2023-08-29
>
> Thanks for your insightful suggestions and we have listed our response to your concerns as follows.
>
> **1.The method is incremental compared to existing methods and the self-attention mechanism is also widely used in other KBQA studies.**
>
> Existing KBQA studies mainly adopt a combination of PLM as text encoder and GNN as graph reasoner to solve the KGQA task. In the two sub-modules, they indeed leverage the self-attention mechanism to understand the complex question or perform message propagation-and-aggregation over the subgraph. While our goal is to provide a new view for modeling the interaction and understanding between text question and graph structured KG, especially for building transferable and powerful KGQA model with efficient training cost. Concretely, we list the following three differences. Concretely, we have the following three differences.
>
> Firstly, inspired by these graph transformer variants, we enable the PLM to simultaneously model question understanding, deep interaction between question and subgraph, and reasoning over subgraph by leveraging adapted subgraph-aware self-attention mechanism.
>
> Secondly, since we do not modify the original model architecture, our approach allows the PLM to utilize pre-trained knowledge to better understand textual relations and entities in the subgraph and question when solving KGQA tasks.
>
> Thirdly, since the PLM is originally trained by learning from general-purpose natural language text, we further propose an adaptation tuning strategy that utilizes a few synthesized data to adapt the PLM to the special input format and attention mechanism. We propose an automatic data construction method for the KGQA task format using LLMs, such as ChatGPT, which has not been explored previously. Our data construction method could inspire the future work about human-free KGQA training data construction with the help of LLMs, since human annotated KGQA datasets are expensive [1].
>
> In addition to these differences, our proposed method also boost the rethinking about the potential of PLM on KGQA tasks even other tasks that require understanding text information and graph structure information simultaneously, such as commonsense question answering [2] or drug modeling with textual description [3].
>
> **2.More case studies and error case studies should be performed to help see the benefit of the proposed method.**
>
> Thanks for your suggestion and we would add more case studies, including error cases after review. Here, we give one case study to see the benefit of our proposed method compared to UniKGQA, a strong baseline which utilizes PLM to model the question and relations and an extra GNN model to perform reasoning over the subgraph. The detailed illustration is as follows, where the predicted reasoning path ("?x" represent the answer entities) is extracted from the reasoning process according to the attention weight:
>
> |      | UniKGQA | ReasoningLM |
> | ----------- | ----------- |----------- |
> | question |    who is queen elizabeth 1 father ?  |   who is queen elizabeth 1 father ? |
> | linked topic entity | "elizabeth i of england" (m.02rg_) |"elizabeth i of england" (m.02rg_) |
> | predicted reasoning path | (m.02rg_, people.person.parents, ?x) | people.person.parents, ?x), (?x, people.person.gender male) |
> | prediction answers | Henry VIII of England (m.03p77), Anne Boleyn (m.09sxn)|Henry VIII of England (m.03p77)|
> |gold answer|Henry VIII of England (m.03p77)|Henry VIII of England (m.03p77)|
>
> As we can see, starting from the topic entity (i.e., m.02rg_), the model needs to find the "people.person.parents" relation to obtain his parents, and then further obtain the farther according to the "people.person.gender" relation and "male" property. However, the GNN module of UniKGQA can only find the "people.person.parents" relation relying on the message propagation-and-aggregation based on the graph structure while fail to attend the further gender property constraint. Therefore, it would return both the mother and father, which is incorrect. In contrast, ReasoningLM can attend the further gender constraint (i.e., (?x, people.person.gender male)) by modeling the question semantics and the connected triples (i.e., (m.02rg_, people.person.parents, ?x), (?x, people.person.gender male) ) in the subgraph within a single PLM. Therefore, our proposed ReasoningLM can better model the interaction between question and subgraph based on the subgraph-aware self-attention mechanism and better understand the question semantics and textual relations and entities in subgraph using the pre-trained knowledge, which is difficult for existing methods that combine a PLM text encoder and a GNN graph reasoner.
>
> Reference:
>
> [1] Yu Gu, Sue Kase, Michelle Vanni, Brian M. Sadler, Percy Liang, Xifeng Yan, Yu Su: Beyond I.I.D.: Three Levels of Generalization for Question Answering on Knowledge Bases. WWW 2021.
>
> [2] Michihiro Yasunaga, Hongyu Ren, Antoine Bosselut, Percy Liang, Jure Leskovec: QA-GNN: Reasoning with Language Models and Knowledge Graphs for Question Answering. NAACL 2021.
>
> [3] Philipp Seidl, Andreu Vall, Sepp Hochreiter, Günter Klambauer: Enhancing Activity Prediction Models in Drug Discovery with the Ability to Understand Human Language. ICML 2023.

---

### Official Review · Reviewer_hcTZ · 2023-08-05

**Soundness:** 3

**Excitement:**

3: Ambivalent: It has merits (e.g., it reports state-of-the-art results, the idea is nice), but there are key weaknesses (e.g., it describes incremental work), and it can significantly benefit from another round of revision. However, I won't object to accepting it if my co-reviewers champion it.

**Paper Topic And Main Contributions:**

The paper introduces ReasoningLM, a pre-trained language model (PLM) that incorporates subgraph reasoning to enhance question understanding and knowledge graph (KG) reasoning. The proposed approach utilizes a breadth-first search (BFS)-based subgraph serialization technique and introduces a subgraph-aware self-attention mechanism within the PLM. An adaptation tuning strategy using ChatGPT is employed to adapt the PLM to the specialized input format and attention mechanism, reducing data construction costs. ReasoningLM is applied to downstream KGQA tasks using parameter-efficient fine-tuning and demonstrates superior performance compared to existing baselines, achieving significant improvements by updating only a fraction of the model's parameters.

**Questions For The Authors:**

1. How will the proposed unified transformer graph encoding method benefit the research community? Can other transformer-based techniques be integrated after this unification? How does this approach differ from previous graph transformers?
2. What is the motivation behind choosing efficient training strategies? How does the performance of ReasoningLM using efficient training compare to full parameter training strategies and existing efficient training methods?

**Reasons To Accept:**

1. The paper presents a novel transformer-only graph encoding method that combines question understanding and KG reasoning.
2. ReasoningLM achieves strong results on multiple KGQA datasets, showcasing its effectiveness.
3. The paper introduces efficient training strategies, which contribute to the model's performance improvements.

**Reasons To Reject:**

1. The motivation behind a unified transformer graph encoding method requires further clarification and justification.
2. The use of sparse masks for graph encoding has been explored in previous works, and the paper does not sufficiently differentiate its approach from those.
3. The rationale behind choosing efficient training strategies is not well-explained, and a comparison with full parameter training strategies and existing efficient training methods would be valuable.

**Reproducibility:**

3: Could reproduce the results with some difficulty. The settings of parameters are underspecified or subjectively determined; the training/evaluation data are not widely available.

**Reviewer Confidence:**

3: Pretty sure, but there's a chance I missed something. Although I have a good feel for this area in general, I did not carefully check the paper's details, e.g., the math, experimental design, or novelty.

---

> ### Author Rebuttal · Authors · 2023-08-29
>
> Thanks for your insightful suggestions and we have listed our response to your concerns as follows.
>
> **1.The motivation behind a unified transformer graph encoding method requires further clarification and justification.**
>
> We have described the motivation for our approach in detail from line 43 to line 96. Here, we want to give a summarization and further clarification. Existing methods for KGQA utilize a PLM encoder such as BERT to attend to the relevant parts of the question, and a GNN reasoner such as GAT to update entity representations based on information from neighboring relations. However, there are two major shortcomings with this approach. First, **the loose integration of PLM and GNN** limits the fine-grained interaction between the question and KG. Second, **the GNN reasoner lacks rich semantic understanding** of relations within subgraphs, particularly for multi-hop relations. Recent work has shown that the transformer architecture of PLMs has the potential to model graph data effectively. Thus, we propose **incorporating the GNN reasoning process directly into the PLM** self to create a more capable PLM that supports both question understanding and structural subgraph reasoning. By doing so, we can enhance the interaction between questions and subgraphs and the subgraph reasoning with better relation semantics understanding. Leveraging the powerful capabilities of PLM, such a unified KGQA model can be effectively and efficiently deployed for different datasets with parameter-efficient fine-tuning, which is superior in realistic scenarios. Our experiments also demonstrate that our model achieves a significant improvement while updating fewer parameters compared to existing complex KGQA models.
>
> **2.The use of sparse masks for graph encoding has been explored in previous works, and the paper does not sufficiently differentiate its approach from those.**
>
> Thanks for your suggestion and we will add this discussion in our paper. Existing sparse attention masks mainly aim to solely model the graph structure by adjusting the transformer architecture. They can not capture fine-grained interactions between the KG and the natural language question simultaneously, making them unsuitable for KGQA task. Besides, as our goal is to build a transferable and powerful KGQA model with efficient training cost, we also devise other new techniques paired with the sparse mask for improving the interaction and understanding between text question and graph structured KG. Concretely, we list the following three differences.
>
> Firstly, we design the special mask to enable the PLM to simultaneously model question understanding, deep interaction between question and subgraph, and reasoning over subgraph by leveraging adapted subgraph-aware self-attention mechanism.
>
> Secondly, since we do not modify the original model architecture, our approach allows the PLM to utilize pre-trained knowledge to better understand textual relations and entities in the subgraph and question when solving KGQA tasks.
>
> Thirdly, we propose an adaptation tuning strategy that utilizes a few synthesized data to adapt the PLM to the special input format and attention mechanism for KGQA task. We propose an automatic data construction method for the KGQA task format using LLMs, such as ChatGPT, which has not been explored previously. Our data construction method could inspire the future work about human-free KGQA training data construction with the help of LLMs, since human annotated KGQA datasets are expensive [1].
>
> **3.How will the proposed unified transformer graph encoding method benefit the research community? Can other transformer-based techniques be integrated after this unification? How does this approach differ from previous graph transformers?**
>
> Our core contribution is to provide a new view for modeling the interaction and understanding between text and graph structure, especially for building transferable and powerful KGQA model with efficient training cost. Previous work adopts a mainstream method that utilizes a PLM as text encoder and a GNN as graph reasoner, and tunes all their parameters for learning the KGQA task. In our work, we find that the PLM can also play the role of GNN and only few synthesized examples can directly adapt it to the KGQA task. Thus, we provide a more unified approach that adapts PLM into GNN with few synthesized data and few tuned parameters leveraging the powerful capability of PLM, which is more efficient than existing work and also achieves the SOTA performance. In addition, our proposed method also boost the rethinking about the potential of PLM on KGQA tasks even other tasks that require understanding text information and graph structure information simultaneously, such as commonsense question answering [2] or drug modeling with textual description [3]. Since our approach does not change the model architecture of the PLM, meaning that it can benefit from the evolvement of advanced PLMs, even popular large language models. We leave the investigation about them as future work.
>
> **4.What is the motivation behind choosing efficient training strategies? How does the performance of ReasoningLM using efficient training compare to full parameter training strategies and existing efficient training methods?**
>
> Across different domains and datasets, the transferability of existing KGQA models is always a challenging problem. Parameter-efficient tunings (e.g., LoRA and Prompt-tuning) have become a hot point for this topic. As only very few task-specific parameters are tuned and most parameters of PLM are shared, it just adds very few cost for the deployment of multiple models for different domains and datasets. In this work, we choose efficient training strategies for such advantage, and our efficient-tuned models have indeed shown its strong performance across different datasets, surpassing SOTA full-parameter tuning models. Our approach also has the following advantages for its good performance. First, we don't modify the architecture of PLM, and thus we can keep its pre-trained knowledge to be utilized and transferred across datasets. Second,  we further propose an adaptation tuning strategy to adapt the original PLM to performing question understanding, interaction between the question and subgraph, and reasoning over the subgraph, controlled by the attention mask. Hence it can be applied to different downstream KGQA tasks with parameter-efficient fine-tuning manner, achieving better performance with only updating a few parameters.
>
> In addition, since we do not modify the original PLM architecture, existing parameter-efficient fine-tuning methods can be seamlessly used. In our paper, we just select one representative adapter tuning [4] method among existing methods. Here, we present additional ablation experiments that compare the performance of full parameter fine-tuning strategy with existing parameter-efficient fine-tuning strategies, including adapter tuning (our used), LoRA [5], (IA)^3 [6], and UniPELT [7]. Our results show that all parameter-efficient fine-tuning methods are able to achieve better results than existing strong baseline (UniKGQA), and their performance is almost indistinguishable from each other. This indicates that our proposed method is compatible with various parameter-efficient fine-tuning methods, and has a good robustness. Although full parameter fine-tuning provides further performance improvements, it needs a cost of updating more than 100 times the number of parameters required by adapter tuning. Such a way enables researchers to trade off the efficiency and effectiveness when employing our approach in real-world applications.
>
> \\begin{array} {|l|c|c|}
> \\hline
> & \\textrm{WebQSP (Hits@1)} & \\textrm{CWQ (Hits@1)} \\\\
> \\hline
> \\textrm{UniKGQA} & 75.1 & 50.7 \\\\
> \\hline
>   \\textrm{Ours w full parameter} & 80.1 & 71.2 \\\\
>   \\textrm{Ours w adapter tuning}  & 78.5 & 69.0 \\\\
>   \\textrm{Ours w LoRA}  & 77.8 & 67.5 \\\\
>   \\textrm{Ours w (IA\)3}  & 78.0 & 68.1 \\\\
>   \\textrm{Ours w UniPELT}  & 78.3 & 69.4 \\\\
> \\hline
> \\end{array}
>
> Reference:
>
> [1] Yu Gu, Sue Kase, Michelle Vanni, Brian M. Sadler, Percy Liang, Xifeng Yan, Yu Su: Beyond I.I.D.: Three Levels of Generalization for Question Answering on Knowledge Bases. WWW 2021.
>
> [2] Michihiro Yasunaga, Hongyu Ren, Antoine Bosselut, Percy Liang, Jure Leskovec: QA-GNN: Reasoning with Language Models and Knowledge Graphs for Question Answering. NAACL 2021.
>
> [3] Philipp Seidl, Andreu Vall, Sepp Hochreiter, Günter Klambauer: Enhancing Activity Prediction Models in Drug Discovery with the Ability to Understand Human Language. ICML 2023.
>
> [4] Neil Houlsby, Andrei Giurgiu, Stanislaw Jastrzebski, Bruna Morrone, Quentin de Laroussilhe, Andrea Gesmundo, Mona Attariyan, Sylvain Gelly: Parameter-Efficient Transfer Learning for NLP. ICML 2019.
>
> [5] Edward J. Hu, Yelong Shen, Phillip Wallis, Zeyuan Allen-Zhu, Yuanzhi Li, Shean Wang, Lu W ang, Weizhu Chen: LoRA: Low-Rank Adaptation of Large Language Models. ICLR 2022.
>
> [6] Haokun Liu, Derek Tam, Mohammed Muqeeth, Jay Mohta, Tenghao Huang, Mohit Bansal, Colin Raffel: Few-Shot Parameter-Efficient Fine-Tuning is Better and Cheaper than In-Context Learning. NeurIPS 2022.
>
> [7] Yuning Mao, Lambert Mathias, Rui Hou, Amjad Almahairi, Hao Ma, Jiawei Han, Scott Yih, Madian Khabsa: UniPELT: A Unified Framework for Parameter-Efficient Language Model Tuning. ACL 2022.

---

### Meta-Review · Area_Chair_xVZd · 2023-09-19

**Recommendation:** 4

**Metareview:**

This paper proposes a new framework that incorporates sub-graph structures into PLMs. In particular, their new framework, ReasoningLM uses subgraph-aware self-attention. Their experimental results show the effectiveness of the proposed method on multiple KBQA. While I agree with the concern about limited novelties noted by all of the reviewers (i.e. sparse masking of transformers has been explored in prior work), incorporating subgraphs for effective masking requires substantial work, and experimental results on KBQA are strong. Therefore, I recommend for acceptance.

---

### Decision · Program_Chairs · 2023-10-07

**Decision:**

Accept-Main

**Comment:**

This paper proposes a new framework that incorporates sub-graph structures into PLMs. In particular, their new framework, ReasoningLM uses subgraph-aware self-attention. Their experimental results show the effectiveness of the proposed method on multiple KBQA. While I agree with the concern about limited novelties noted by all of the reviewers (i.e. sparse masking of transformers has been explored in prior work), incorporating subgraphs for effective masking requires substantial work, and experimental results on KBQA are strong. Therefore, I recommend for acceptance.